# Accelerated Mobilization of Organic Carbon from Retrogressive Thaw Slumps on the Northern Taymyr Peninsula

Philipp Bernhard[1], Simon Zwieback[2], and Irena Hajnsek[1,3]

[1]Institute of Environmental Engineering, ETH Zurich, 8093 Zurich, Switzerland ETH Zürich
[2]Geophysical Institute, University of Alaska Fairbanks, Fairbanks, AK 99775 USA
[3]Microwaves and Radar Institute, German Aerospace Center (DLR) e.V., 82234 Wessling, Germany

**Correspondence:** Philipp Bernhard (bernhard@ifu.baug.ethz.ch)

**Abstract.** With climate change, Arctic hillslopes above ice-rich permafrost are vulnerable to enhanced mass wasting and organic carbon mobilization. In this study we use TanDEM-X-derived digital elevation models to document an approximately 43-fold increase in thaw slumping and concomitant 28-fold increase in carbon mobilization on the Northern Taymyr Peninsula from 2010 to 2021. The available observations allowed us to compare two time-periods, from 2010/11 to 2016/17 and from 2017/18 to 2020/21, and contrast Retrogressive Thaw Slump (RTS) activity between these periods. We find that that all quantities describing RTS activity increased in the observed period. The total volumetric change per year increased from about $0.17 \cdot 10^6 \, \mathrm{m}^3 \, \mathrm{yr}^{-1}$ to $7.4 \cdot 10^6 \, \mathrm{m}^3 \, \mathrm{yr}^{-1}$, a 43-fold increase. The observed surge in RTS activity is mainly driven by the initiation of new RTS, indicated by the 17-fold increase in active RTS numbers from 82 to 1404 and the relative low average volumetric change rate per RTS increase of 2.3. In annual Sentinel-2 imagery, the number of detected RTSs in a subregion increased 10-fold in 2020. This coincides with a severe heatwave that occurred in northern Siberia in 2020. The area-to-volume scaling of the RTSs varied only slightly over time, despite the 2020 heatwave, indicating a robustness of the relationship to such an event. To estimate the slump-mobilized organic carbon, we intersected the elevation changes with a soil organic carbon (SOC) map, with contrasting assumptions about the deep carbon pool and massive ice content. We estimated that the SOC mobilization rate increase 28-fold. The normalization of the SOC mobilization rate to our study region yields values of $11 \, \mathrm{gC} \, \mathrm{yr}^{-1} \, \mathrm{m}^{-2}$ with a confidence interval of 5 to $38 \, \mathrm{gC} \, \mathrm{yr}^{-1} \, \mathrm{m}^{-2}$. Comparison to an independent estimate of the Net Ecosystem Exchange of $4.1 \pm 13.0 \, \mathrm{gC} \, \mathrm{yr}^{-1} \, \mathrm{m}^{-2}$ illustrates the importance of RTS activity to the carbon cycle. These results underscore that mass wasting is an important but commonly neglected component of the Arctic carbon cycle, and particularly sensitive to extreme events.

# 1   Introduction

In the northern Hemisphere about 15 % of the landmass is underlain by permafrost (Obu, 2021). With increasing temperatures these permafrost regions become vulnerable to thaw, which can occur in a gradual, slow mode or more rapid and regionally localized (Kokelj and Jorgenson, 2013; Schuur et al., 2015). Permafrost thaw has major impacts on the local ecosystem and can also influence the global climate by the mobilization of organic carbon that is stored in the ground. This mobilization can lead to strong positive feedback effects with dramatic consequences for the global climate (Schuur et al., 2008; Grosse et al., 2011). The intensity and timing of the permafrost carbon contribution to the global budget is not well constrained and has large uncertainties (López-Blanco et al., 2019; Zscheischler et al., 2017). Current Earth system models focus on large scale landscape changes and the gradual thaw of permafrost (Lawrence et al., 2015; McGuire et al., 2018). This neglects spatially discrete and comparatively rapid processes that occur in ice-rich permafrost regions, termed thermokarst. Thermokarst features develop and evolve on manifold scales, both in time and in extent. Turetsky et al. (2020) modelled the effect of climate change on these thermokarst landscapes and identified a large potential of future carbon release by the initiation of new features as well as the continued evolution of present ones. Nevertheless, strong model assumptions and limited data availability lead to large uncertainties, especially in terms of the most abrupt thaw, were single initiation events, like high summer temperatures can lead to widespread initiations (Lewkowicz and Way, 2019). To date, there is only one study estimating organic carbon mobilized by rapid permafrost mass wasting on a regional scale (Ramage et al., 2017), but no satellite-derived estimates are available.

In this work, we will focus on one form of hillslope thermokarst, namely retrogressive thaw slumps (RTS) also termed thermo-cirques or cryogenic landslides (Lantuit and Pollard, 2005; Leibman et al., 2014). They are characterized by a steep headwall and a scare zone where the thawed material from the headwall is transported downslope. RTSs initiate through the exposure of ice-rich permafrost by the removal of the protective active layer. The reason for this can be manifold and depend on the landscape settings and processes. Along coasts or rivers, mechanical erosion is the main driver of RTS initiation (Burn and Lewkowicz, 1990; Kokelj et al., 2015). On hillslopes, high summer temperatures and strong precipitation events can lead to active layer detachments due to high pore water pressure resulting from low hydraulic conductivity and which can then further develop into RTSs (Jorgenson and Osterkamp, 2005; Lewkowicz, 2007; Lamoureux and Lafreniere, 2009; Lewkowicz and Way, 2019; Jones et al., 2019). RTSs expand upslope due to the continual exposure and melt of ground ice at a headwall, thus mobilizing thawed materials which are transported downslope through the scar zone (Kokelj and Jorgenson, 2013; Zwieback et al., 2020). RTSs can grow where ground ice content and topographic settings allows for a continued instability and removal of thawed soils (Burn and Lewkowicz, 1990; Lacelle et al., 2010; Kokelj and Jorgenson, 2013). On a pan-Arctic scale, RTSs have a large variation in size, ranging from small slumps with headwall heights of less than a meter up to large mega slumps with headwalls heights of up to 40 m (Swanson and Nolan, 2018; Kokelj et al., 2015; Murton et al., 2017). Also the retreat rates of the headwall are variable and can reach values of up to several tens of meter per year (Kokelj and Jorgenson, 2013; Kokelj et al., 2015; Lacelle et al., 2015). Past RTS studies have shown that the prevalence, geomorphic characteristics and carbon mobilization are related to soil properties, ice contents and topography which vary across the pan-Arctic landscape, highlight-

ing the need for large-scale satellite-based monitoring (Lantz and Kokelj, 2008; Kokelj and Jorgenson, 2013; Zwieback et al., 2020).

Satellite remote sensing is a promising avenue for the monitoring of thaw slump activity and carbon mobilization. In recent years, RTS analysis studies based on optical remote sensing techniques found an increase in the number as well as sizes of RTSs (Lantz and Kokelj, 2008; Ramage et al., 2017; Nitze et al., 2018; Lewkowicz and Way, 2019; Huang et al., 2020; Runge et al., 2022). Here, time-series of optical satellite images are used to detect active RTSs by identifying disturbed soil due to the movement of the headwall and the transport of sediments downslope. Another approach to analyse RTSs, is based on differencing digital elevation models (DEMs) that are obtained at different times to measure the induced volumetric change due to the retreating headwall (Bernhard et al., 2020). This approach is more direct since the immediate retreat of the headwall and the resulting mobilization of soil is used as the proxy for RTS activity. In this study we use DEMs generated from single-pass Interferometric Synthetic Aperture Radar (InSAR) observations acquired by the TanDEM-X satellites (Krieger et al., 2007) to map and investigate RTSs. The spatial resolution of about $10\,\mathrm{m}$ and vertical height resolutions of the order of about 1 to $2\,\mathrm{m}$ is high enough to be sensitive to most RTSs. Bernhard et al. (2022) showed that with this approach RTS activity can be described in form of various scaling laws known from landslide studies in temperate climate region, making it possible to compare RTS activity between Arctic region, but the temporal variability in these laws are unknown. In recent years several improved soil organic carbon (SOC) maps covering the whole Arctic were released (Hugelius et al., 2014; Mishra et al., 2021) which in combination with the quantification of the volumetric change rates potentially allows to make estimates of the amount of mobilized SOC due to RTS activity.

In this work our goal is to map and investigate RTSs on the Northern Taymyr Peninsula, a region containing massive ground ice, remnant from the Kara Ice-Sheet and which is known to be susceptible to thaw slumping (Grosval'd et al., 1986; Yershov, E.D., 1989; Alexanderson et al., 2002). The available observations allows us to investigate and contrast two time periods. Our analysis includes the quantification of the induced volumetric and area change rates and obtaining different scaling laws, namely the volume to area scaling and the probability density functions. We then estimate the amount of mobilized SOC due to RTS activity based on a ground model including massive ice contents and a pan-Arctic SOC map. Additionally, we use higher temporal resolution optical satellite data on a small part in our study region to assess the impact of a severe heatwave that started in the beginning of 2020 and ended in summer (Overland and Wang, 2021).

The main objectives of this work are:

1. Quantify the RTS activity based on the area and volumetric changes using elevation models generated from the TanDEM-X observations in two time-periods (2010/11/12 to 2016/17 and 2017/18 to 2020/21)

2. Measure the change in RTSs areas and volumes including the RTS scaling relations and its change over time

3. Estimate of the SOC content that was released due to RTS activity based on the induced volumetric changes and a comparison to other carbon exchange studies in the Arctic permafrost regions

4. Assess the relation between RTS initiation and the 2020 Siberian heatwave using annual Sentinel-2 observations

## 2 Study Area

The study region is located on the Northern Taymyr Peninsula in Siberia (Russia) and spans from 75°N to 77.5°N in latitude and 88°E to 106°E in longitude (Figure 1a). We focus on a study region of $68\,000\,\text{km}^2$ in the time period from 2010/11 to 2016/17, but limited data availability constrained us to a smaller subset of about $27\,000\,\text{km}^2$ in the time period from 2017/18 to 2020/21. To the south, the study region is limited by the Byrranga Mountains and to the North by the Kara Sea. The relief is low to moderate, with the exception of the Byrranga Mountains to the south which reaches an elevation of about 400m above sea level. The study region is classified as a graminoid tundra (Walker et al., 2003). In the most northern part the vegetation is moist with low-growing plants of mostly grasses, forbs and mosses. Further to the south the vegetation changes towards a dwarf-shrub and forb tundra (Walker et al., 2003).

The climate is characterised as polar Arctic with a mean annual air temperature of about -10°C. The mean July air temperatures are 2 to 5 °C in the far North and 7 to 10 °C in the South of the study region. During winter the region experiences monthly mean air temperatures below -20 °C (Figure 1b). The area is underlain by continuous cold permafrost, with estimated permafrost temperatures of -11°C to -8°C (Obu et al., 2018).

Quaternary glaciations and marine transgressions have made a mark on the topography and stratigraphy of the Northern Taymyr Peninsula. The entire peninsula was covered by the Kara Sea Ice Sheet during the Saalian glacial period (140 - 130 ka; MIS 6; Hjort et al. (2002)). Deglaciation during the Eemian interglacial (MIS 5e) was followed by renewed glaciation during the Early Weichselian in MIS 5d to 5a (Hjort et al., 2004; Batchelor et al., 2019). During the Middle and Late Weichselian (MIS 4 to 2) the ice retreated step-wise and also temporary re-advanced leading to several ice-marginal zones (Northern Taymyr ice-marginal zones - NTZ) north of the Byrranga Mountains with associated features of glacial, glaciofluvial and glaciolacustrine deposits including buried glacial ice. The Geocryology of the USSR also indicate some marine deposits in the region (Proskurnin et al., 2016). Two ice-marginal zones (NTZ 1 and 2) have been associated with the Early and Middle Weichselian and are located close to each other. The most recent of these ice-marginal zones stemmed from the Last Glacial Maximum (NTZ 3) (Möller et al., 2011). The location of the ice-marginal zones are shown in Figure 1a.

## 3 Methods

### 3.1 DEM generation from TanDEM-X observations

To map RTS-induced volume changes and carbon mobilization, we differenced repeat DEMs generated from interferometric SAR (InSAR) TanDEM-X observations and identified RTS locations. In total, our dataset contains 991 bistatic single-pol TanDEM-X observations. The final DEMs have a planimetric resolution of 10 to 12 m and vertical accuracies below 2 m in areas with high coherences. We used a standard approach to generate DEMs from InSAR data and used the TanDEM-X 12 m DEM product as reference DEM to reduce unwrapping errors and facilitate orthorectification and tilt removal. We only used observation taken during the winter months, where a frozen landscape can be expected, since otherwise sizeable errors are present in the DEM difference images (Bernhard et al., 2020).

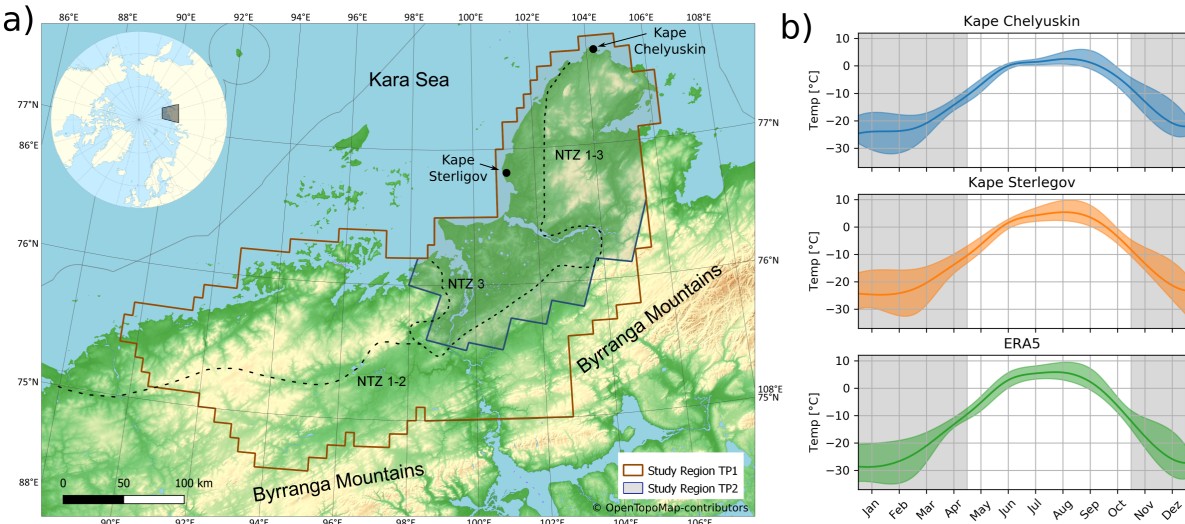

**Figure 1.** a) Map of the Northern Taymyr Peninsula with the study regions of time-period 1 (TP1) and time-period 2 (TP2). The Northern Taymyr Ice-Marginal Zones (NTZ 1-3) are shown as dotted line and are taken from Möller et al. (2011). Basemap: OpenTopoMap (open-topomap.org). b) Mean annual air temperatures for the average over the study time period from 2010 to 2021 with temperature data from the weather stations at Kape Chelyuskin at 77°43'N, 104°18'E (rp5.ru, 2022a) and Kape Sterlegov at 75°24'N, 88°47'E (rp5.ru, 2022b) as well as averaged ERA5 air temperature data combined over the total study region (Muñoz Sabater, 2019). The boarders of the coloured area around the mean values indicate the minimum and maximum monthly temperature values in the study time period. The darker region indicate winter months from November to April.

The temporal coverage allowed us to determine DEMs for the winters 2010/11, 11/12, 16/17, 17/18 and 2020/21. Data for 2020/21 are restricted to only a part of the Northern Taymyr Peninsula (See Figure 1). Furthermore, acquisitions before 2017 are obtained in an ascending orbit, whereas after 2017 only observations from a descending orbit are available. Comparing DEMs obtained from different orbits is prone to substantial errors in rugged terrain, associated with layover, shadow, and imperfect co-registration. We thus generated DEM difference images for two time periods: 2010/11/12 to 2016/17 (TP1), and 2017/18 to 2020/21 (TP2). The expected vertical accuracies can be characterised by the distance between the satellite pair and converted to the Height of Ambiguity (HoA), defined as the elevation change per interferometric phase cycle. We estimated the standard deviation based on the HoA and the interferometric coherence, a measure of the phase quality (Martone et al., 2012). In the available time series of observations, the HoA varies between 30 m and 80 m, corresponding to an estimated standard deviation of 1 m to 1.8 m at a coherence of 0.9 (Figure 2).

### 3.2 RTS detection, property extraction and error calculation

To identify and delineate active RTSs in the DEM difference images we used the semi-automated method presented in Bernhard et al. (2020) and Bernhard et al. (2022). This approach works as follow: First significant elevation changes are detected using a

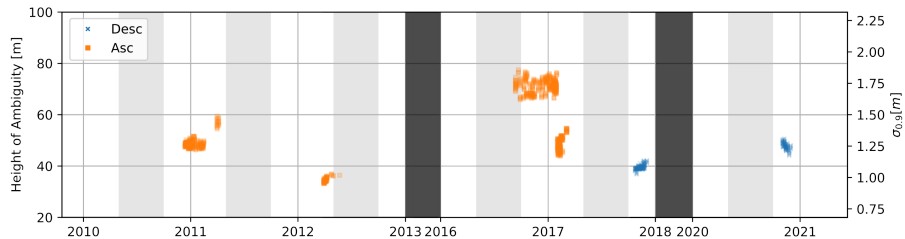

**Figure 2.** TanDEM-X observations over time with Height of Ambiguity on the left and the computed standard deviation with an assumed coherence of 0.9 on the right. The period from winter 2010/11, winter 2011/12 to winter 2016/17 is Time-Period 1 (TP1), the observations from winter 2018/19 to winter 2020/21 is time-period 2 (TP2). In winter 2010/11, winter 2011/12 and winter 2016/17 only ascending (Asc) observations are available, in winter 2018/19 and winter 2020/21 only descending (Desc) observations are available. The grey areas indicate the months from November to April where we can expect a frozen landscape.

multi-scale blob detection approach. Secondly, false positives are discarded by manual inspection, retaining only RTSs. Lastly, the area affected by RTSs is delineated by the generation of polygons. In the manual inspection of the detection location, as well as in the delineation of the affected area, we additionally used time-series of Sentinel-2 and Planet Rapid-Eye images (Drusch et al., 2012; Planet-Team, 2018). For each RTS we computed the volumetric and area change based on the drawn

polygons. To simplify the analysis we normalized the properties to changes per year.

To estimate the uncertainty of the volumetric changes, we accounted for multiple sources of uncertainty. The error in elevation is directly related to the noise in the interferometric phases and can be estimated using the coherence $\gamma$, a measure of decorrelation and thus for the interferometric phase quality. During winter, systematic errors from dry snow are negligible and the main contribution to the decorrelation is due to low backscatter intensities (low signal-to-noise ratio) (Krieger et al., 2007; Rizzoli

et al., 2017; Bernhard et al., 2020). The coherence estimate can be translated into an estimate on the elevation error using the Cramer-Rao bound:

$$\sigma_H(\gamma, \mathrm{HoA}, L) = \frac{\sqrt{1-\gamma^2}}{\gamma\sqrt{2L}} \frac{\mathrm{HoA}}{2\pi} \tag{1}$$

where HoA is the Height of Ambiguity, defined as the height change corresponding to one interferometric phase cycle and $L$ is the number of looks to reduce speckle noise (Kay, 1993; Krieger et al., 2007).

The estimated elevation error together with an error in the area allows to compute the final error on the volumetric change by:

$$\sigma_V^{+,-} = H_{\mathrm{RTS}} \cdot A_{\mathrm{RTS}} \cdot \sqrt{\left(\frac{\sigma_H}{H_{\mathrm{RTS}}}\right)^2 + \left(\frac{\sigma_A^{+,-}}{A_{\mathrm{RTS}}}\right)^2} \tag{2}$$

where $A_{RTS}$ is the area affected by RTS retreat, $H_{RTS}$ is the averaged elevations changes of all pixels ($n_{pix}$)

$$H_{\mathrm{RTS}} = \frac{\sum_i^{n_{\mathrm{pix}}} h^i}{n_{\mathrm{pix}}} \tag{3}$$

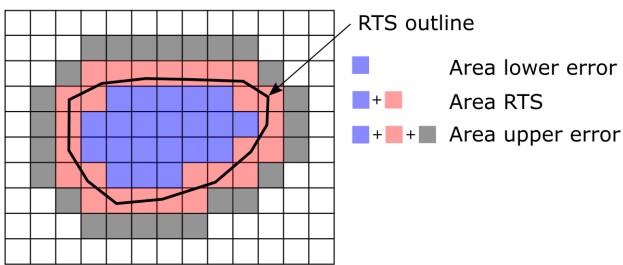

**Figure 3.** Schematic of the area error calculation of an RTSs. The lower error is defined by only using pixels when an erosion operation in applied. The RTS area is computed by taking all pixels that are inside and touched by the drawn polygon. The upper error is computed using the RTS pixels and apply a dilation operation.

$\sigma_H$ is the error on the averaged elevations

$$\sigma_H = \frac{\sqrt{\sum_i^{n_{\mathrm{pix}}}(\sigma_h^i)^2}}{n_{\mathrm{pix}}} \tag{4}$$

and $\sigma_A^{+,-}$ the upper and lower error in the RTS area. To estimate the error in the area component we employ an approach based on morphological operations (Figure 3). First we extract the pixels that are covered by the drawn polygon. To estimate the upper bound we apply a morphological dilation, that increases the size by one pixel on the outside and compute the area difference ($\sigma_A^+$). The lower bound is estimated in a similar way by applying an erosion operation ($\sigma_A^-$). This approach likely overestimates the error but other approaches like the drawing of polygons by several trained persons was not feasible.

To compute the uncertainties in the ratios between the obtained quantities in TP1 and TP2 we assumed no correlation and used error propagation.

### 3.3 RTS scaling relations

To quantify the relationship between area and volumetric change rates we assumed an anisotropic scaling with an exponent $\alpha$, that relates the area and volume change rates by $V \approx A^\alpha$. We used an orthogonal distance regression model to fit a straight line in log space since both variables are affected by measurement errors (Boggs and Rogers, 1990; Markovsky and Van Huffel, 2007). We calculated the goodness of the fit using the RMSE, $R^2$ and p-value (in log-space).

To quantify the change in RTS activity in a probabilistic way we studied the probability density function (PDF) of the volumetric and area change rates per year. The PDF is defined as

$$p(Q_{\mathrm{RTS}}) = \frac{1}{N_{\mathrm{RTS}}} \frac{\delta N_{\mathrm{RTS}}}{\delta Q_{\mathrm{RTS}}} \tag{5}$$

where $Q_{\mathrm{RTS}}$ indicates the volumetric change, respectively, the area change of RTSs per year, $N_{\mathrm{RTS}}$ the total number of RTS in the inventory, $\delta N_{\mathrm{RTS}}$ the number of RTS with affected areas between $Q_{\mathrm{RTS}}$ and $Q_{\mathrm{RTS}} + \delta Q_{RTS}$ and $\delta Q_{\mathrm{RTS}}$ the bin widths. We then fitted a three parameter Inverse Gamma Function to the sampled data points (Malamud et al., 2004). To estimate the error of the fit we used a bootstrap algorithm (Ohtani, 2000). We analysed the PDF using three quantities: the rollover point,

defined as the peak in the PDF which corresponds to the most common occurrence in the distribution, the cutoff-point after
which the distribution starts to follow a power law, and the power law scaling parameter $\beta$ describing this power-law. For the
computation of the rollover point we identified the peak in the fitted Inverse Gamma Function. For determining the cutoff value
and the exponential scaling exponent we used the method of Clauset et al. (2009) and quantified the uncertainty again using a
bootstrap algorithm.

## 3.4  Mapping RTS-induced organic carbon mobilization

To estimate the amount of carbon that is mobilized due to RTS activity we intersected the elevation change estimates with the
currently most accurate soil organic carbon (SOC) map available for our study region (Mishra et al., 2021). However, this data
set is poorly constrained in less studied regions such as the Northern Taymyr Peninsula. In addition, it only contains data for
the uppermost 3 m. Furthermore, RTSs are known to develop at location with abundant massive ice, of which no datasets are
available for our study region.

To account for these limitations we employed the following approach: For the estimation of the SOC in the uppermost 3 m we
use the SOC map by Mishra et al. (2021) (SOC-shallow). In the SOC map separate values for the 0 - 1 m, 1 - 2 m and 2 - 3 m
depths are available. To estimate the SOC content in the deeper layers (SOC-deep), we employ two models. In the first model
(SOC-M1) we fit a linear function to the SOC values in the upper columns and interpolate to the deeper layers. We computed
SOC-deep by integrating the fitted line from 3 m depth till it reaches zero or the depth of the measured elevation change is
reached. In a second model we used an exponential decay function ($\beta e^{-\alpha}$) to estimate SOC-deep (SOC-M2). In this model
the SOC values do not reach zero in deeper layers. See Figure 4a for a schematic comparison of the two models. A decreasing
SOC content is consistent with data of deep carbon measurements from other regions in the Arctic (Strauss et al., 2017).

Additionally to the SOC, the massive ice content needs to be known. Here the data availability is more scars and uncertain than
it is for the SOC content. Yershov, E.D. (1989) estimated massive ice-content on the Northern Taymyr Peninsula in the range
from 30% to 70%. Other studies that estimated massive ice content at RTS locations have found values in a wide range. Couture
and Pollard (1998) found a massive ice content of 30% in an RTS in Eureka down to a depth of 5.6 m and Pollard (1990) found
massive ice content of up to 85% in RTSs on Herschel Island (both northern Canada). Our study region is part of the North
Taymyr Ice-Marginal zone and it is likely that most RTSs are located in moraine areas with buried glacial ice which suggests a
high ice content (Yershov, E.D., 1989; Alexanderson et al., 2002). We built two additional models for the massive-ice content
in the following way: We assumed an overburden of 1 m above massive ice, based on field observations by Alexanderson et al.
(2002). For the deeper layers we employ two models: Ground-Ice Model 1 (GI-M1) with a massive ice content of 40% and
Ground-Ice Model 2 (GI-M2) with a massive ice content of 80%. See Figure 4b for a schematic visualization of our model
assumptions.

To compute the final SOC mobilization we then add SOC-shallow and SOC-deep, considering the different model assumptions.
To calculate the amount of SOC that is mobilized per RTS we employ the following equation:

$$\text{SOC}_{\text{RTS}} = \sum_{i}^{n_{\text{pix}}} \left( \text{SOC}_{>1\text{m}}(h_i) + \text{SOC}_{<1\text{m}}(h_i) \cdot (1 - \text{GI}) \right) \cdot A^{\text{pix}} \tag{6}$$

where $h_i$ is the depth of pixel $i$, $A^{\mathrm{pix}}$ is the area per pixel, $\mathrm{SOC}_{<1\mathrm{m}}$ is the SOC content per pixel in the upper most meter, $\mathrm{SOC}_{>1\mathrm{m}}$ the SOC content from $1\,\mathrm{m}$ down to a depth of $h$ considering SOC Model 1, respectively, 2 and GI returns the percentage of massive ice for models 1, respectively 2.

To estimate the error of the mobilized carbon, we consider both, the error in the SOC values as well as in the area and volume. For the error in the SOC values we used the error provided by Mishra et al. (2021) for each layer in the upper most 3 m. For the deeper layers we extrapolate the error of the 2 to 3 m range. To account for the error in area and height we computed a lower and upper error by using the lower respective upper area bound and then subtract respectively add the standard deviation of the elevation measurements to the final depth.

To gauge the importance of slump-induced carbon mobilization relative to the net ecosystem exchange (NEE), we compared our results with a recent upscaling study by Virkkala et al. (2021). This study modelled NEE rates across the Arctic based on field data. We used the data available for our study site and computed the mean and standard deviation from the provided data for the period 1995 to 2015 and for the year 2015 for the comparison with our results. Comparing the NEE rates directly to our data is challenging due to the complexity of carbon mobilization pathways as well as the diverse nature of RTS activity regarding timing and spatial variability. To nevertheless show the importance of carbon mobilization due to RTS activity, we use the total amount of mobilized SOC per year in our study region and normalise the mobilization rate to the study region size of TP2 with the removal of open sea areas. The size of this region amounts to $25.96 \cdot 10^9\,\mathrm{m}^2$. Since RTS activity shows strong clustering in specific regions due to the dependency on soil properties like massive ice content, this rate depends strongly on the region used for the normalization which should be taken into consideration when interpreting the result. Since the difference between SOC-M1 and SOC-M2 is small, we used the average of the models and compute two SOC mobilization rates per unit area for the two ground-ice models with the associated error ranges for each time period.

## 3.5 Relationship between RTS initiation and the 2020 Siberian heatwave using annual Sentinel-2 observations

To relate RTS activity to meteorological drivers, we compiled an annual RTS inventory from Sentinel-2 data for a small subregion of about $370\,\mathrm{km}^2$ and associated RTS initiation with historic climate data. The climate data are based on the ECMWF ERA5 hourly data with a spatial resolution of $0.1°$, from which we obtained the 2 m temperature data (Muñoz Sabater, 2019). To calculate the average values per day we used all grid values covering our study region. From the daily mean temperatures we then derived the thawing degree days (TDD) and freezing degree days (FDD) (Boyd, 1976).

To track the formation of new RTS, we compiled an annual inventory of RTSs. The short summer season and high cloud cover limited that amount of usable data. We generated and downloaded one cloud- and snow-free image per year and as late as possible in the summer season. The mosaics were derived for the years 2016 to 2020 at a resolution of 10 m (Bands 2, 3, 4, 8) in the Google Earth Engine (Gorelick et al., 2017). We scanned each obtained image for thaw slump activity and if such activity was visible, we drew a polygon outlining the disturbed area. Note that this area is different than the area indicating elevation loss in the DEM difference images. Furthermore, since RTSs form during summer, the first year a RTS is visible does not necessarily correspond to the year of initiation, since the growth rate needs to be large enough to be visible in the 10 m resolution optical images.

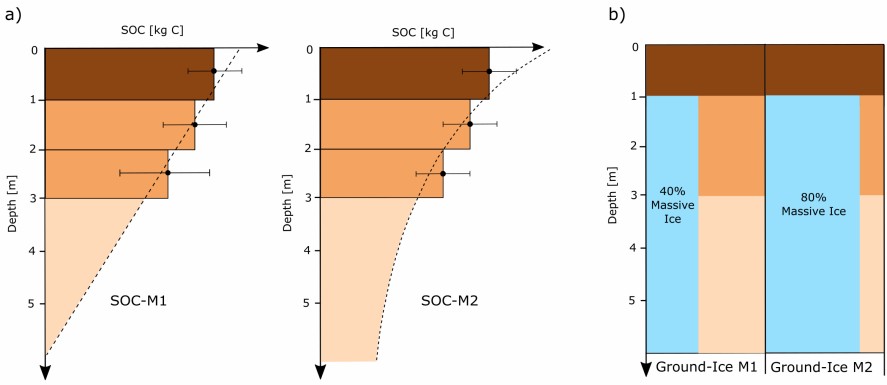

**Figure 4.** Schematics of the developed models. a) shows the model assumptions to compute SOC-deep and associated SOC values for SOC-M1 and SOC-M2. For SOC-M1 we fit a linear function to the SOC values in the upper 3m to estimate SOC-deep. For SOC-M2 we fit an exponential function to estimate SOC-deep. The final values of SOC-M1 and SOC-M2 are then the sum of all layers. b) shows the two assumptions for the massive ground ice content with no massive ice in the upper 1m and below a massive ice content of 40% for Ground-Ice Model 1 and 80% for Ground-Ice Model 2.

## 4 Results

We investigated the RTS activity on the Northern Taymyr Peninsula over two time periods and computed the area and volumetric change rates for each RTS. In the following paragraphs we will present (1) an overview of the general RTS activity with special emphasis on the change between the two time-periods; (2) the fitted probability density functions with the estimation of the rollover, cutoff and exponential decay components as well as the estimated area-to-volume scaling laws; (3) an estimation of the mobilized SOC mobilization rates and an analyse of the influence of our SOC model assumptions; and (4) an investigation of the relationship between the Siberian heatwave in 2020 and the RTS activity using a mapping approach based on Sentinel-2 data.

### 4.1 RTS activity in TP1 and TP2

The number of detected active RTSs increased from 82 in TP1 to 1404 in TP2, corresponding to a 17-fold increase. We can observe that most RTS activities are located close to the North Taymyr Ice-Marginal Zone (Figure 5). The study region size of TP1 is much larger than for TP2 due to the limitations in the available TanDEM-X observation in winter 2020/21. Nevertheless, the RTS activity in TP1 outside the study region of TP2 was very low with only 6 RTSs located outside this study region. For the following comparison we only used RTSs inside the TP2 study region.

The total volumetric and area change rates increased from the first to the second time period (Figure 6). The total volumetric change per year increased from about $0.17$ to $7.4 \cdot 10^6 \, \text{m}^3 \, \text{yr}^{-1}$ corresponding to a 43-fold increase. The computed errors span over large ranges around these values. Similar to the total volumetric change rates, the total area change rate increased 56-fold from TP1 to TP2. When normalized by the number of RTSs, the volumetric change rate per RTS increased from about

$2.3 \cdot 10^3 \, \mathrm{m}^3 \, \mathrm{yr}^{-1} \, \mathrm{RTS}^{-1}$ to $5.3 \cdot 10^3 \, \mathrm{m}^3 \, \mathrm{yr}^{-1} \, \mathrm{RTS}^{-1}$ corresponding to a 2.3-fold increase. For the average area change rate we found a 3-fold increase. All values can be seen in Table 1.

We also investigated the subset of RTSs that were active in TP1. Of the 82 RTSs in TP1, 57 showed a continued growth, 19 stabilized (not detectable in TP2) and 6 were outside of the spatial coverage of TP2. The average area and volume change rates increased for the subset of the 57 RTSs stronger that for the total population of RTSs: by a factor of 3.9 in volumetric change rate and by a factor of 4.5 in the area change rate. The main factor of the strong increase in the total volumetric and area change rates is due to the increase in RTS number and not due to the enlargement of existing RTSs, although the increase in RTS growth rates of existing RTSs was larger than for the newly detected ones.

## 4.2 RTS scaling relations

The area-to-volume conversion factor, corresponding to the slope of the fitted lines in Figure 7a, decreased slightly from TP1 to TP2. However, this decrease was within the estimated error range. We estimated the one and two sigma prediction intervals and found that when using an area measurement of RTS change to estimate the volumetric change the expected error correspond to about 14% to 16% of the volume for one sigma and about 40% to 45% for a two sigma prediction interval.

The distributions of the area and volume probability density distributions shifted towards higher values from TP1 to TP2 (Figure 7b and 7c). The fit of an inverse gamma function for TP2 was good with a $R^2$ values of 0.98 for the volumetric change rates and 0.97 for the area change rates. For TP1 the fit performed worse ($R^2_{\mathrm{area}} = 0.84$ and $R^2_{\mathrm{volume}} = 0.90$). This is likely related to the low number of RTSs in TP1. The PDFs can be characterised by the rollover and cutoff locations. The increase in these quantities was similar to the average change rates per RTS with increase ratios based on the volumetric change of 1.7 for the rollover and 2.9 for the cutoff. Similarly the increase ratios based on the area change was 2.9 for the rollover and 4.4 for the cutoff. The exponential decay coefficient decreased slightly for both, area and volumetric change rate PDFs, by $2.25 \pm 0.14$ to $1.76 \pm 0.13$ (volume) and $3.02 \pm 0.37$ to $1.97 \pm 0.20$ (area). The biggest difference between the two time periods was that the exponential decay part continues for both PDFs to about an order of magnitude larger values in TP2. All values can be seen in Table 1.

## 4.3 RTS-induced organic carbon mobilization

The estimated SOC mobilization rates based on the two SOC and massive ice assumptions can be seen in Figure 8a with the resulting TP2 to TP1 ratios in Figure 8b. For the SOC model with an exponential decreasing SOC value in the deep layers (SOC-M2) and 40% massive-ice (GI-M1) we obtained the largest SOC mobilization rates of $13.8 \cdot 10^6 \, \mathrm{kgC} \, \mathrm{yr}^{-1}$ for TP1 which increased to $378.5 \cdot 10^6 \, \mathrm{kgC} \, \mathrm{yr}^{-1}$ in TP2. This increase corresponds to an about 27-fold increase in total SOC mobilization. The difference to the linearly decreasing SOC model for the deep layer (SOC-M1) is small, with $0.5 \cdot 10^6 \, \mathrm{kgC} \, \mathrm{yr}^{-1}$ reduction for TP1 and $8.5 \cdot 10^6 \, \mathrm{kgC} \, \mathrm{yr}^{-1}$ for TP2. For GI-M2 with 80% massive ice in the layers below 1 m the SOC mobilization rates were smaller with $9.9 \cdot 10^6 \, \mathrm{kgC} \, \mathrm{yr}^{-1}$ in TP1 and $284.0 \cdot 10^6 \, \mathrm{kgC} \, \mathrm{yr}^{-1}$ for TP2 using SOC-M2 (29-fold increase). Here the difference to the SOC-M1 mobilization rates are even smaller with $0.2 \cdot 10^6 \, \mathrm{kgC} \, \mathrm{yr}^{-1}$ reduction for TP1 and $2.8 \cdot 10^6 \, \mathrm{kgC} \, \mathrm{yr}^{-1}$ for TP2. The mobilized carbon predominantly originates from the upper most meter in our estimates. For both time periods the yearly

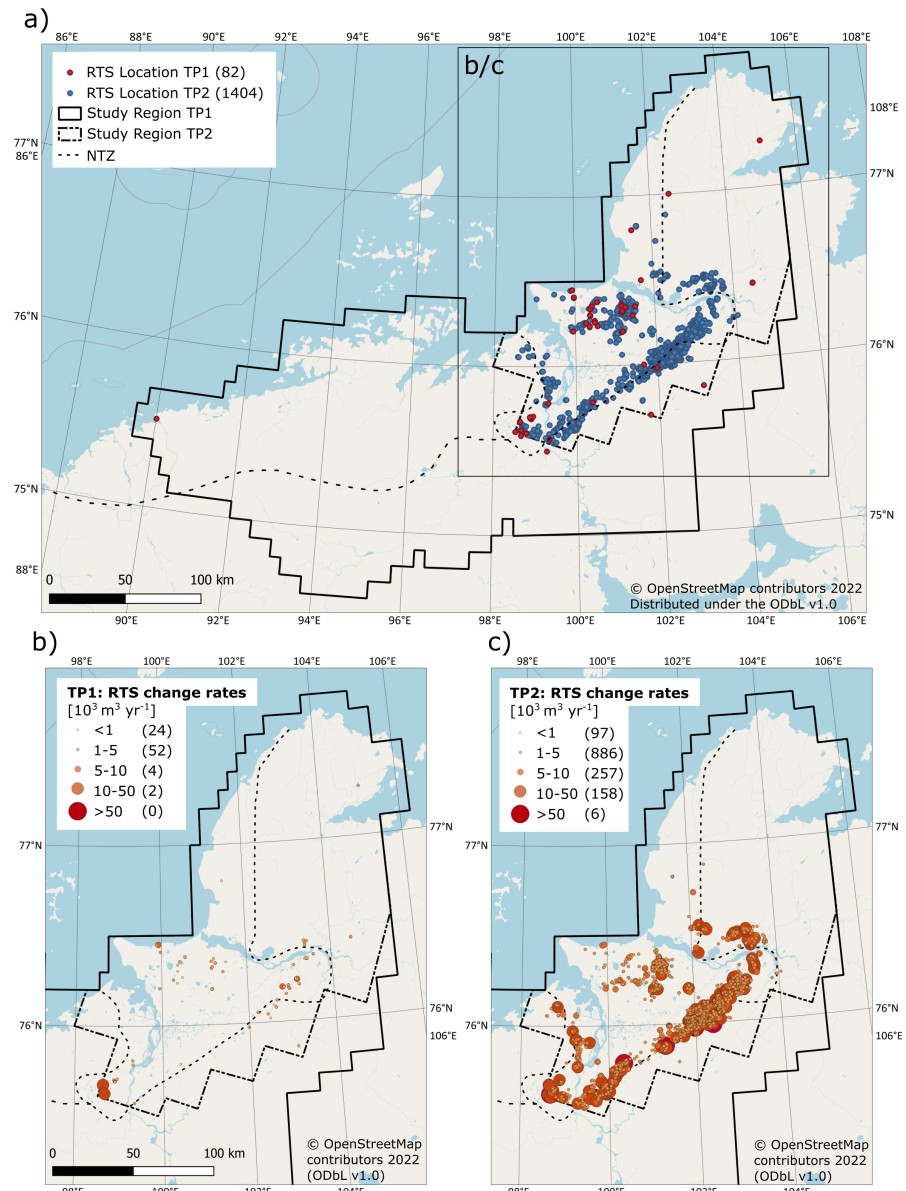

**Figure 5.** Overview of the RTS location and change rates in the study region. In brackets we show the number of RTSs in each class. a) shows the total study region of TP1 and TP2 with the RTS location. b) shows the yearly volumetric RTS change rates in TP1 zoomed in to the study region of TP2. c) shows the yearly volumetric RTS change rates in TP2 in the study region of TP2. Basemap: © OpenStreetMap contributors 2022. Distributed under the Open Data Commons Open Database License (ODbL) v1.0

volumetric changes rates of this upper most meter contributed about 30% to 35% of the total. Translating the volumetric changes to the associated SOC mobilization, the contribution of the upper most meter is about 60% for GI-M1 and 85% for

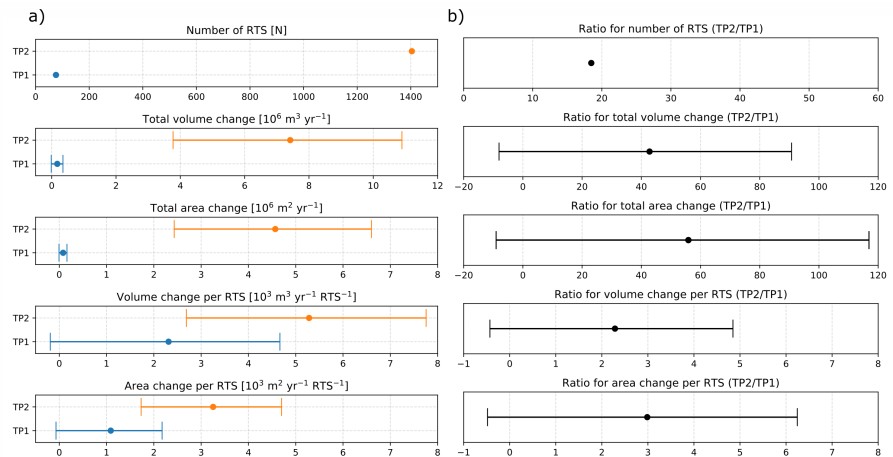

**Figure 6.** Increase rates from TP1 to TP2. a) shows the number of RTS, the total yearly volumetric change rates, the total yearly area change rates, the yearly volumetric change rates per RTS, and the yearly area change rates per RTS. In b) the ratios between TP1 and TP2 are shown.

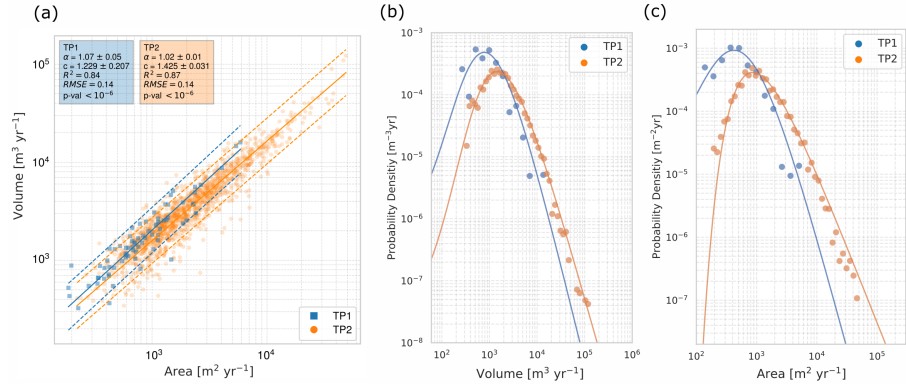

**Figure 7.** a) shows the area to volume scaling relation and obtained fitting parameters. The dashed line shows the 95% prediction interval. b) shows the PDFs with fitted inverse gamma function of the yearly area change rates. c) shows the PDFs with fitted inverse gamma function of the yearly volumetric change rates.

GI-M2. The deep layers below 3 m contribute about 15% to 20% to the total volumetric change and again computing the associated SOC mobilization the contribution is about 5% to 10% of the total. Nevertheless, SOC-M2 mobilizes about 40% to 50% more SOC in the deep layers as SOC-M1 (Figure 9).

The error ranges in our estimated SOC mobilization rates are large, spanning ranges that decrease or increases the SOC mobilization rates by factors of 3 to 4. These large uncertainties arise due to the combination of errors in the area and volumetric change rates of the mapped RTSs (DEM-related) as well as from the uncertainties in the SOC maps (SOC-related). The contribution of DEM-related error are generally larger than the SOC-related error spanning from 1.5 to 2 times larger to up to

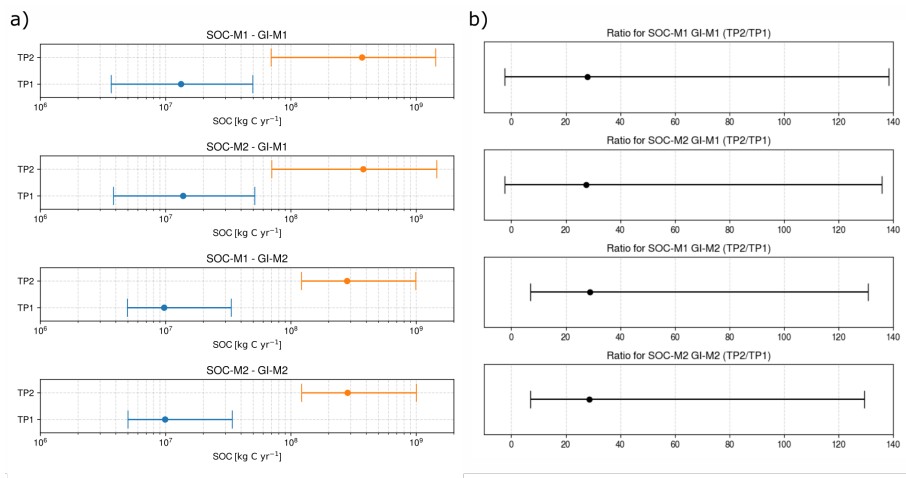

**Figure 8.** a) shows the SOC mobilization rates of RTSs for all model combinations in TP1 and TP2. A 26- to 28-fold increase in mobilization rates is visible. In b) the ratios between TP1 and TP2 are shown.

8 to 9 times larger error depending on the model assumptions (Table 2).

The slump-driven OC mobilization rates are of comparable magnitude to the region's NEE estimated by Virkkala et al. (2021) (Figure 10). For TP1 we find SOC mobilization rates per unit area of $0.52_{0.14}^{1.95}\,\mathrm{gC\,yr^{-1}\,m^{-2}}$ (GI-M1) and $0.38_{0.19}^{1.30}\,\mathrm{gC\,yr^{-1}\,m^{-2}}$

(GI-M2). These rates increase to $14.42_{2.69}^{55.70}\,\mathrm{gC\,yr^{-1}\,m^{-2}}$ (GI-M1) and $10.88_{4.68}^{38.44}\,\mathrm{gC\,yr^{-1}\,m^{-2}}$ (GI-M2). Using the data from Virkkala et al. (2021) we estimated yearly NEE rates averaged over our study region of $4.1\pm13.0\mathrm{gC\,yr^{-1}\,m^{-2}}$ using the averaged data for the 1990 to 2015 period and $10.3\pm12.4\mathrm{gC\,yr^{-1}\,m^{-2}}$ in 2015. A comparison of our SOC mobilization rates and to the study by Virkkala et al. (2021) can be seen in Figure 10.

## 4.4 Annual analysis of RTS initiation

93% of the new RTSs were identified in the year 2020 in the Sentinel-2 imagery, corresponding to the exceptionally warm year (Figure 11). The number of RTSs in the Sentinel-2 study region in 2016 was 8 and we did not find any newly initiating RTSs in the years 2017 and 2018. In 2019 several new RTSs initiated increasing the number of active RTSs to 21. A further strong increase in 2020 increased the number of RTSs to 270 (Figure 11b). Even if some increase in RTS activity occurred in 2019, most RTS initiation and growth happened in the summer of 2020. This is also consistent with the high TDDs values during

2020 related to a Siberian heatwave (Figure 11c). The preceding winter of 2019/20 was also exceptionally warm.

We additionally compared the Sentinel-2 to the TanDEM-X mapped RTSs and noted all RTSs in the TanDEM-X sample that could be related to Sentinel-2 mapped RTSs. Here we found that all 8 RTSs that were active in 2016 and the following years were also detected by the TanDEM-X approach. For the 13 RTSs that initiated in 2019, only 2 could be related to RTS location in the TanDEM-X RTS dataset (10/21, 47.6%). For 2020, 122 of the total 270 (45.1%) Sentinel-2 RTSs could be associated

with TanDEM-X detected RTSs.

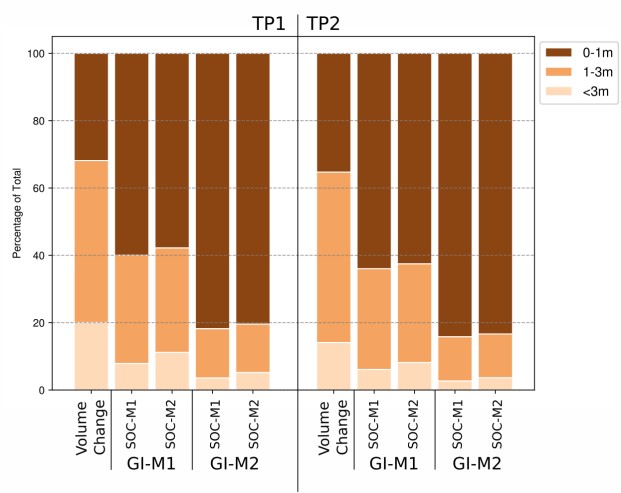

**Figure 9.** Mobilization of Volume and SOC separated by different depth columns in terms of percentage of total.

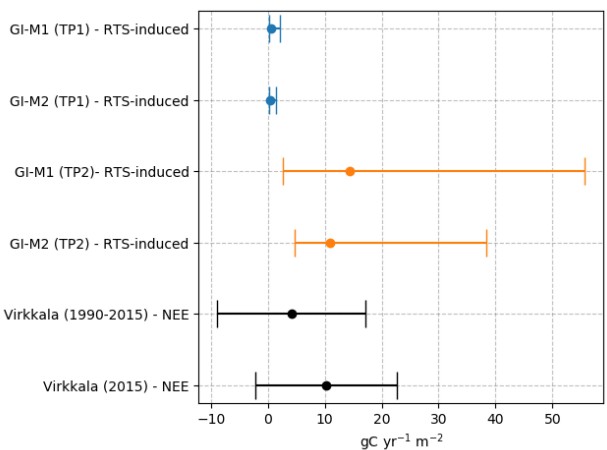

**Figure 10.** SOC mobilization rates per unit area and comparison to independent study by Virkkala et al. (2021) estimating Net Ecosystem Exchange (NEE) rates in our study region.

## 5 Discussion

### 5.1 Acceleration of RTS activity on the Northern Taymyr Peninsula

Our regional satellite-based assessment revealed a large increase in the number of RTSs and the associated total volumetric change rates. The number of RTSs increased 17-fold from the first time period from 2010/11 to 2016/17 to the second from 2017/18 to 2020/21, while the volumetric change rate increased 40-fold. The landscape change was thus driven primarily by

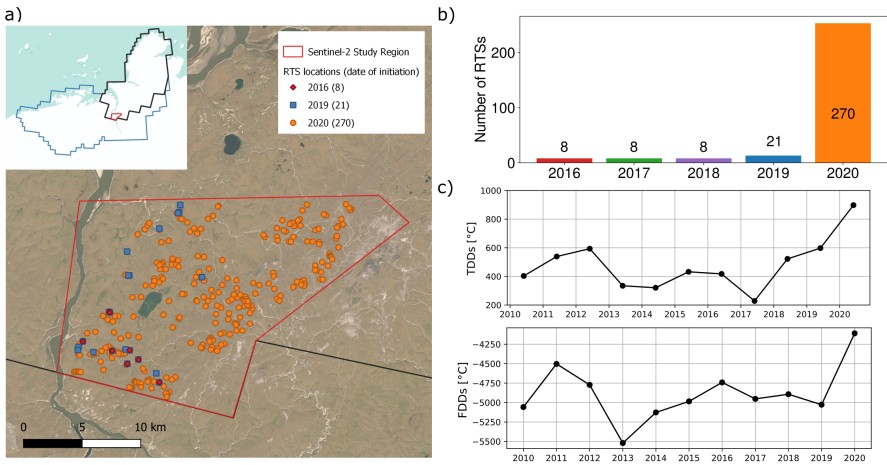

**Figure 11.** a) shows an overview of the Sentinel-2 study region and location of the mapped RTS color coded by the year of initiation. b) shows the number of mapped RTSs per year. c) shows the TDDs (summed over each year) and FDDs (summed over winter period) from 2010 to 2020. The marker location corresponds to the first of June of each year for the TDDs and first of January corresponding to that winter for the TDDs.

**Table 1.** All computed quantities for TP1 and TP2 and their ratios including the computed error.

| Quantity | TP1 | TP2 | Ratio (TP2 to TP1) |
|---|---|---|---|
| Number of RTS [N] | 76 | 1404 | 17.1 |
| Total Volume [$10^6\,\mathrm{m}^3\,\mathrm{yr}^{-1}$] | $0.17^{0.34}_{-0.02}$ | $7.42^{10.89}_{3.77}$ | $42.78^{90.74}_{-8.05}$ |
| Total Area [$10^6\,\mathrm{m}^2\,\mathrm{yr}^{-1}$] | $0.08^{0.16}_{-0.01}$ | $4.57^{6.60}_{2.43}$ | $55.89^{116.89}_{-9.03}$ |
| Volume (RTS) [$10^3\,\mathrm{m}^3\,\mathrm{yr}^{-1}\,\mathrm{RTS}^{-1}$] | $2.31^{4.66}_{-0.19}$ | $5.28^{7.75}_{2.69}$ | $2.32^{4.85}_{-0.43}$ |
| Area (RTS) [$10^3\,\mathrm{m}^2\,\mathrm{yr}^{-1}\,\mathrm{RTS}^{-1}$] | $1.08^{2.17}_{-0.07}$ | $3.25^{4.70}_{1.73}$ | $3.01^{6.24}_{-0.48}$ |
| Rollover (Volume) [$10^2\,\mathrm{m}^3\,\mathrm{yr}^{-1}$] | $6.85^{7.31}_{6.39}$ | $11.53^{12.21}_{10.92}$ | $1.68^{1.86}_{1.51}$ |
| Cutoff (Volume) [$10^3\,\mathrm{m}^3\,\mathrm{yr}^{-1}$] | $3.54^{4.09}_{3.02}$ | $10.35^{12.86}_{7.84}$ | $2.92^{5.11}_{0.73}$ |
| Rollover (Area) [$10^2\,\mathrm{m}^3\,\mathrm{yr}^{-1}$] | $3.86^{4.24}_{3.48}$ | $11.32^{12.45}_{10.19}$ | $2.93^{3.84}_{2.03}$ |
| Cutoff (Area) [$10^3\,\mathrm{m}^3\,\mathrm{yr}^{-1}$] | $1.63^{1.90}_{1.37}$ | $7.12^{8.90}_{5.34}$ | $4.37^{9.26}_{-0.53}$ |
| SOC-M1 GI-M1 [$10^6\,\mathrm{kgC}\,\mathrm{yr}^{-1}$] | $13.28^{49.67}_{3.68}$ | $370.03^{1431.43}_{69.51}$ | $27.86^{138.86}_{-2.43}$ |
| SOC-M2 GI-M1 [$10^6\,\mathrm{kgC}\,\mathrm{yr}^{-1}$] | $13.80^{51.49}_{3.83}$ | $378.48^{1480.57}_{70.38}$ | $27.43^{135.86}_{-2.42}$ |
| SOC-M1 GI-M2 [$10^6\,\mathrm{kgC}\,\mathrm{yr}^{-1}$] | $9.73^{33.41}_{4.96}$ | $281.13^{993.18}_{121.82}$ | $28.88^{129.31}_{7.01}$ |
| SOC-M2 GI-M2 [$10^6\,\mathrm{kgC}\,\mathrm{yr}^{-1}$] | $9.91^{34.02}_{5.01}$ | $283.95^{1002.89}_{121.82}$ | $28.66^{129.31}_{7.01}$ |
| SOC per unit area GI-M1 [$\mathrm{gC}\,\mathrm{yr}^{-1}\,\mathrm{m}^{-2}$] | $0.52^{1.95}_{0.14}$ | $14.42^{55.70}_{2.69}$ | $27.64^{137.10}_{-2.43}$ |
| SOC per unit area GI-M2 [$\mathrm{gC}\,\mathrm{yr}^{-1}\,\mathrm{m}^{-2}$] | $0.38^{1.30}_{0.19}$ | $10.88^{28.44}_{4.69}$ | $28.77^{129.79}_{7.10}$ |

**Table 2.** SOC mobilization rate error contribution separated in DEM-related and SOC-related. The units are: $10^6 \, \mathrm{kgC\,yr^{-1}}$.

| Model and TP | DEM-related lower error | DEM-related upper error | SOC related error | DEM (lower)/ SOC error | DEM (upper)/ SOC error |
|---|---|---|---|---|---|
| SOC-M1, GI-M1 (TP1) | 9.39 | 18.94 | 5.54 | 1.69 | 3.42 |
| SOC-M2, GI-M1 (TP1) | 9.54 | 19.69 | 5.55 | 1.72 | 3.55 |
| SOC-M1, GI-M2 (TP1) | 6.98 | 13.05 | 1.65 | 4.24 | 7.92 |
| SOC-M2, GI-M2 (TP1) | 7.03 | 13.30 | 1.65 | 4.26 | 8.08 |
| SOC-M1, GI-M1 (TP2) | 214.77 | 546.12 | 145.25 | 1.48 | 3.76 |
| SOC-M2, GI-M1 (TP2) | 215.68 | 558.30 | 145.30 | 1.48 | 3.84 |
| SOC-M1, GI-M2 (TP2) | 165.46 | 386.99 | 43.93 | 3.77 | 8.81 |
| SOC-M2, GI-M2 (TP2) | 165.76 | 391.05 | 43.95 | 3.77 | 8.90 |

RTS (re-)initiation, although the volumetric change per RTS also increased somewhat by a factor of two.

The exceptionally warm year of 2020 coincided with a more than 10-fold increase in the number of RTSs visible in the Sentinel-2 images. 2020 was characterized by a very warm winter and a record warm summer, with TDDs twice as high as on average. Mass initiations of RTSs following extreme summer temperatures have been documented on the Canadian Arctic Archipelagos (Jones et al., 2019; Lewkowicz and Way, 2019) and the Yamal Peninsula in Siberia (Khomutov et al., 2017). This study is the first that identified such an initiation event for RTSs on the Northern Taymyr Peninsula. Similar to the Canadian Arctic Archipelago, the thin organic layers, fine-grained sediments and near-surface massive ground ice render hillslopes susceptible to shallow slope failures. These in turn develop into RTSs as ground ice is continually exposed. The inventoried RTSs first detected in 2020 (Figure 11a) are on regular hillslopes and not adjacent to water bodies. Taken together, these observations indicate that ice-rich hillslopes in cold permafrost are particularly sensitive to extremely warm temperatures.

Most RTSs in our sample are located close to the Northern Taymyr Ice-Marginal zone of the Wechsilian Last Glacial Maximum (NTZ 3), identified by Möller et al. (2011). The region is characterized by glacial and glaciofluvial surface features and often contain buried relic glacial ice. This finding agrees with a previous study by Kokelj et al. (2017) in northern Canada where RTSs are found at the maximum and recessional positions of the Laurentide Ice Sheet.

## 5.2 RTS scaling relations and their response to a strong initiation event

The area-to-volume scaling of the RTSs varied slightly over time, despite the 2020 heatwave, indicating a robustness of the relationship to such an event. The area-to-volume scaling coefficient $\alpha$ was low throughout, with an alpha of about 1 corresponding to RTSs whose growth is predominantly driven by an increase in area. The smaller $\alpha$ estimate of 1.02 in TP2 could be associated with the predominance of juvenile RTSs formed in 2020. Furthermore, RTSs in our study region are predominantly shallow and elongated, in contrast to RTSs with large headwalls elsewhere (Bernhard et al., 2022). For landslides studies

different scaling coefficients for different terrain and soil types as well as landslide sizes have been proposed to reduce errors when estimating volumetric change rates from area changes (Larsen et al., 2010; Chen et al., 2019). Such an approach should also be considered for RTS area-to-volume conversions.

On the other hand, the probability density distributions described by the rollover and cutoff values shifted to larger values, similar to the increase in the average RTS growth rates. Additionally, the exponential decay part of the distribution extended to an order of magnitude larger RTSs in area and volumetric change rates. This indicates that with climate warming the distribution of RTS change rates can shift towards larger values and is not stable inside a region.

## 5.3 Substantial organic carbon mobilization from RTSs

Our novel satellite-based assessment of RTS-induced organic carbon mobilization revealed a substantial acceleration during the study period. The estimated carbon mobilization per year increased approximately 28-fold from the first to the second time period. This acceleration was predominantly due to the large number of new RTSs in the second period, with most new RTSs being detected after the 2020 heatwave. The complex non-linear response of the Arctic carbon cycle to summer temperatures necessitates regular satellite-based monitoring.

While the acceleration in carbon mobilization is clearly evident, the magnitude of the rates is uncertain due to a multiple of error sources, like the strong modelling assumptions and observational errors. The first assumption, SOC content below <3 m, had a limited impact on the estimated mobilization because the mobilization of shallow materials was dominant (Figure 9). Nevertheless, the exponential decay model mobilized about double the amount of SOC in these deep layers compared to the linear model. Our assumption for the massive-ice content with 40% and 80% massive ice below 1 m, showed larger variabilities of about 30% to 40% in the total SOC mobilization rates. The largest error was found to be due to observational uncertainties in the volume changes measurements. Additional TanDEM-X observations as well as combinations of different data-sources like optical RTS mapping or the ArcticDEM have the potential to decrease this DEM-related error contribution.

Additional to the errors related to the modelling assumption and errors in the estimated volumetric change rates, several other error sources are present which are difficult to quantify. The comparison to the Sentinel-2 mapped RTSs has shown, that about 50% of RTSs are missed in the TanDEM-X mapping approach. This is likely due to small headwall heights and relative recent initiations in summer 2020. But the missed RTSs can potentially mobilize a significant amount of organic carbon, due to typically larger soil organic carbon contents in the upper soil layer. On the other hand, RTS re-initiation can lead to an overestimation of the amount of mobilized carbon, since the upper soil layer with high carbon contents has already been mobilized. Regarding mapping errors, we tried to estimate the error in the area change by increasing and decreasing the size of the drawn polygons. Additional human errors in drawing the polygons are possible and not included in the error estimation. An approach taken by previous studies (e.g. Lewkowicz and Way (2019)) were polygons were drawn multiple times by different trained persons was not feasible. Furthermore, relating the changes to individual RTSs becomes difficult for RTSs in close proximities and due to RTS coalescence. In this study we separated RTSs based on the induced elevation changes. RTSs that seem connected based on the induced vegetation changes obtained from optical and infrared observations could thus be related to multiple RTSs in our RTS inventory. This can occur if these RTSs are only connected by small or slow moving headwalls or

by the flow of the thawed soil downwards.

Our mobilization estimates show that RTSs are an important part of the carbon cycle on regional scales. The mobilized organic carbon is of at least the same order of magnitude as the NEE, when normalized by the total area. It is to note, that in this study we only estimated the amount mobilized carbon. The fate of this mobilized carbon is uncertain and depends strongly on its decomposability and the general landscape setting (Cassidy et al., 2017; Bröder et al., 2021). The timing and amount

of greenhouse gases released from RTS-mobilized organic carbon is thus difficult to quantify (Vonk and Gustafsson, 2013; Abbott and Jones, 2015; Turetsky et al., 2020). Slump-induced mobilization can nevertheless greatly affect the overall carbon balance of a region, even if only a part of the mobilized carbon becomes part of the active carbon cycle. Our estimation of the large scale carbon mobilization rates due to RTS activity is a first step to a better quantification of the impact of degrading permafrost on the permafrost carbon feedback.

## 5.4   Towards Pan-Arctic monitoring

SOC mobilization due to thermokarst development and more specifically due to RTS activity are not included in current global climate models. Turetsky et al. (2020) has estimated that hillslope thermokarst features can contribute up to 20% of the general permafrost carbon release in the future. With this study we have further confirmed that the carbon mobilization potential due to RTSs can be significant and occur in a non-linear, rapid way. This rapid mobilization and the strong spatial variability of RTS

activity across the Arctic highlights the importance of observing RTS mobilization rates in the future.

Accurate elevation measurements are important for pan-Arctic monitoring, but the available TanDEM-X observations have limitations. A major limitation in our dataset is due to the disparate look directions (ascending and descending passes), which change the spatial ground resolution related to aspect in presence of topography. The emerging uncertainties can be mitigated by more frequent observations in variable orbit passes. Additional to the spatial resolution, the vertical height resolution is

important and mainly depends on the distance between the satellites, characterized by the Height of Ambiguity. Observations with small Height of Ambiguities have the disadvantage of impeded phase unwrapping, but due to relatively small variation in the topography, the unwrapping procedure is manageable and such observations can reduce the vertical height accuracy significantly. Future single-pass InSAR missions may provide more accurate and frequent elevation data at a higher resolution. Additionally, other DEM datasets like the ArcticDEM can improve the RTS volumetric change estimates (Morin et al., 2016).

Satellite-derived carbon mobilization estimates require accurate SOC and massive ice content products. More accurate SOC and massive ice content data with accurate error estimates, specifically for RTS locations are highly desirable. RTS locations are special in the regard that they develop in high ground ice settings. Furthermore, accurate estimates of the deep carbon are important when applying our methodology on regions containing RTSs with large headwall heights. Furthermore, models predicting greenhouse gas release from SOC mobilization rates are needed to quantify the impact of the hillslope thermokarst

contribution to climate change.

# 6 Conclusions

Elevation change estimates from TanDEM-X observations reveal a substantial acceleration of RTS activity and mobilized carbon on the Northern Taymyr Peninsula in Siberia between 2010 to 2021. We found that the number of RTSs and volumetric and area changes increased substantially with for example a 40-fold increase in the volumetric change rates. We attribute the increase to the mass initiation of new RTSs during the 2020 heatwave, based on Sentinel-2 image analysis over a small subregion.

An approximately 28-fold increase in the mobilization of organic carbon accompanied the increase in RTS activity. This first satellite-driven regional assessment was based, in addition to the TanDEM-X elevation changes, on a soil organic carbon map, assumptions about the soil organic carbon at depth below 3 m and two critical assumptions about the massive ice content. Our sensitivity analyses indicates that the influence of the assumptions on the estimates is small compared to the overall 28-fold increase in mobilization rates. On regional scales, the large mobilization from winter 2017/18 to winter 2020/21 is of at least the same magnitude as the estimated Net Ecosystem Exchange.

Our findings show that hillslope thermokarst can be a major, but largely neglected component of the Arctic carbon cycle. The amount of carbon mobilized by RTSs responded sharply and non-linearly to warming, underscoring the sensitivity of upland landscapes underlain by cold ice-rich permafrost to increasing temperatures. While the fate of the mobilized carbon (mineralization, burial) remains poorly constrained, the magnitude of the flux necessitates its regular monitoring, inclusion in Arctic carbon budgets and incorporation into land surface models. Satellite remote sensing will be an indispensable tool for monitoring carbon mobilization by permafrost mass wasting across the Arctic.

*Data availability.* Locations, polygones and extracted properties of RTSs are available at: https://doi.org/10.3929/ethz-b-000529493. Sentinel-2 are available from the Copernicus Open Access Hub (https://scihub.copernicus.eu). TanDEM-X CoSSC data are not freely available but can be requested from the German Aerospace Center (DLR) and accessed through the EOWEB (https://eoweb.dlr.de)

*Author contributions.* PB conducted the DEM processing, the manual RTS mapping, analysed the data and drafted the initial manuscript, SZ provided critical guidance and contributed to the writing of the manuscript, IH provided guidance and corrections to the final manuscript.

*Competing interests.* The authors declare no competing interests.

*Acknowledgements.* The authors are very thankful to the two reviewers, Nina Nesterova and Justine Ramage, who were essential in improving the manuscript and providing insightful comment and suggestions. The authors would also like to thank Gustav Hugelius and Umakant

Mishra for a discussion and provision of the soil organic carbon map, the German Aerospace Center for the provision of the data takes from the TanDEM-X mission as well as the European Space Agency for the Sentinel-2 data provision.

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
