# Peer review of "Accelerated Mobilization of Organic Carbon from Retrogressive Thaw Slumps on the Northern Taymyr Peninsula"

_The Cryosphere, 2022_

## Referee Comment (RC2)

**1075 and 1074 polygons. Screenshots of Planet September 2020**

---

## Author Response (AR1)

We thank the reviewer for the detailed and constructive comments. We considered each comment carefully and address them point by point below. We hope that our modifications will make the manuscript clearer and more complete.

In the following we abbreviate: **RC1** (Reviewer 1 Comment), **RC2** (Reviewer 2 Comment) and **AC** (Author Comment)

**Reviewer 1:**

**Introduction**

**RC1:** L34: Ramage et al., 2017 is the wrong reference. I guess you want to refer to Ramage, J.L., Irrgang, A.M., Morgenstern, A. and Lantuit, H., 2018. Increasing coastal slump activity impacts the release of sediment and organic carbon into the Arctic Ocean. Biogeosciences, 15(5), pp.1483-1495.

**AC:** We changed the reference to Ramage et al. 2018

**RC1**: L38-39: I would suggest expending on a few more reasons explaining their expansion

**AC:** We expanded the paragraphs and added more details:

*In this work, we will focus on one form of hillslope thermokarst, namely retrogressive thaw slumps (RTS) also termed thermocirques or cryogenic landslides (Lantuit and Pollard, 2005; Leibman et al., 2014). They are characterized by a steep headwall and a scare zone where the thawed material from the headwall is transported downslope. RTSs initiate through the exposure of ice-rich permafrost by the removal of the protective active layer. The reason for this can be manifold and depend on the landscape settings and processes. Along coasts or rivers, mechanical erosion is the main driver of RTS initiation (Burn and Lewkowicz, 1990; Kokelj et al., 2015). On hillslopes, high summer temperatures and strong precipitation events can lead to active layer detachments due to high pore water pressure resulting from low hydraulic conductivity and which can then further develop into RTSs (Jorgenson and Osterkamp, 2005; Lewkowicz, 2007; Lamoureux and Lafreniere, 2009; Lewkowicz and Way, 2019; Jones et al., 2019). RTSs expand upslope due to the continual exposure and melt of ground ice at a headwall, thus mobilizing thawed materials which are transported downslope through the scar zone (Kokelj and Jorgenson, 2013; Zwieback et al., 2020). RTSs can grow where ground ice content and topographic settings allows for a continued instability and removal of thawed soils (Burn and Lewkowicz, 1990; Lacelle et al., 2010; Kokelj and Jorgenson, 2013).*

**RC1:** L43-44: this sentence is unclear and needs to be rewritten

**AC:** We rewrote the sentence to:

*Past RTS studies have shown that the prevalence, geomorphic characteristics and carbon mobilization are related to soil properties, ice contents and topography which vary across the pan-Arctic landscape, highlighting the need for large-scale satellite-based monitoring (Lantz and Kokelj, 2008; Kokelj and Jorgenson, 2013; Zwieback et al.,2020).*

**RC1:** L47: please add reference: Ramage, J. L., Irrgang, A. M., Herzschuh, U., Morgenstern, A., Couture, N., and Lantuit, H. (2017), Terrain controls on the occurrence of coastal retrogressive thaw slumps along the Yukon Coast, Canada, J. Geophys. Res. Earth Surf., 122, 1619– 1634, doi:10.1002/2017JF004231.

**AC:** We added the reference

**RC1:** L57: Do you mean "between Arctic regions"? Or did you forget to mention the region with which RTSs from the arctic region can be compared?

**AC:** We mean "between Arctic regions", for example the scaling coefficients relating the area change to the volumetric change vary between Arctic regions (e.g. Banks Island vs Yamal/Gydan).

**RC1**: L62: please provide references to "a region that is known to be susceptible to thaw slumping".

**AC:** We added the reference and slightly rewrote the sentence:

*In this work our goal is to map and investigate RTSs on the northern Taymyr Peninsula, a region containing massive ground ice, remnant from the Kara Ice-Sheet and which is known to be susceptible to thaw slumping (Grosval'd et al., 1986; Yershov, E.D., 1989; Alexanderson et al., 2002).*

**RC1:** L69: please repeat which periods are considered

**AC:** We added the time periods

**RC1:** L71-72: I find this terminology quite complex and I do not understand what you mean by "probability density function". I suggest you to simplify the methodology e.g "measure the change in RTSs areas and volumes"

**AC:** We changes item two of the objectives to:

*2. Measure the change in RTSs areas and volumes including the RTS scaling relations and its change over time*

**RC1:** L73: replace "an estimation" by "estimate"

**AC:** Corrected

**RC1:** L77: change "our study region" by "the study region"

**AC:** Corrected

**RC1:** Technical -- Figure 7. a) show the Area to volume scaling relation and obtained fitting parameter.

**AC:** Corrected

**Discussion:**

Substantial organic carbon mobilization from RTSs: you mention that the landscape change is mostly driven by RTSs were re-initiating. The sediments that are remobilizing might have lower carbon content since part of it was already mobilized. How do you think that this re-initiation affects your estimates of carbon mobilization? I suggest adding a few sentences on this. There are a few studies on carbon mobilization on stabilized and re-initialized RTS that you could use:

Cassidy, A.E., Christen, A. and Henry, G.H., 2017. Impacts of active retrogressive thaw slumps on vegetation, soil, and net ecosystem exchange of carbon dioxide in the Canadian High Arctic. *Arctic Science*, *3*(2), pp.179-202.

Bröder, L., Keskitalo, K., Zolkos, S., Shakil, S., Tank, S.E., Kokelj, S.V., Tesi, T., Van Dongen, B.E., Haghipour, N., Eglinton, T.I. and Vonk, J.E., 2021. Preferential export of permafrost-derived organic matter as retrogressive thaw slumping intensifies. *Environmental Research Letters*, *16*(5), p.054059.

Abbott, B.W. and Jones, J.B., 2015. Permafrost collapse alters soil carbon stocks, respiration, CH 4, and N2O in upland tundra. *Global Change Biology*, *21*(12), pp.4570-4587.

**AC:** We addressed the points mentioned in the discussion:

*The comparison the Sentinel-2 mapped RTSs has shown, that about 50\% of RTSs are missed in the TanDEM-X mapping approach. This is likely due to small headwall heights and relative recent initiations in summer 2020. But the missed RTSs can potentially mobilize a significant amount of organic carbon, due to typically larger soil organic carbon contents in the upper soil layer. Furthermore, RTS re-initiation can lead to an overestimation of the amount of mobilized carbon, since the upper soil layer with high carbon contents has already been mobilized. [...]*

*Our mobilization estimates show that RTSs are an important part of the carbon cycle on regional scales. The mobilized organic carbon is of at least the same order of magnitude as the NEE, when normalized by the total area. It is to note, that in this study we only estimated the amount mobilized*

*carbon. The fate of this mobilized carbon is unknown and depends strongly on its decomposability and the general landscape setting. The timing and amount of greenhouse gases released from RTS mobilized organic carbon is thus difficult to quantify (Vonk and Gustafsson, 2013; Abbott and Jones, 2015; Turetsky et al.,2020). Slump-induced mobilization can nevertheless greatly affect the overall carbon balance of a region, even if only a part of the mobilized carbon becomes part of the ecosystem carbon fluxes (Cassidy et al., 2017; Bröder et al., 2021). Our estimations of large scale carbon mobilization rates is a first step to better quantify the impact of degrading permafrost on the permafrost carbon feedback.*

**Reviewer 2:**

**RC2:** Lines 36-41*: Please, define the term RTS in your study more specifically. What do you include when saying RTS? For example, H. Lantuit and W. H. Pollard, 2005, draw a scheme where characteristics of RTS include the presence of a headwall, slump floor, mudlobe etc. So, I'd suggest a bit more of description and mechanisms behind RTS occurrence.

**AC:** W added more details in the describtion of RTSs:

*In this work, we will focus on one form of hillslope thermokarst, namely retrogressive thaw slumps (RTS) also termed thermocirques or cryogenic landslides (Lantuit and Pollard, 2005; Leibman et al., 2014). They are characterized by a steep headwall and a scare zone where the thawed material from the headwall is transported downslope. RTSs initiate through the exposure of ice-rich permafrost by the removal of the protective active layer. The reason for this can be manifold and depend on the landscape settings and processes. Along coasts or rivers, mechanical erosion is the main driver of RTS initiation (Burn and Lewkowicz, 1990; Kokelj et al., 2015). On hillslopes, high summer temperatures and strong precipitation events can lead to active layer detachments due to high pore water pressure resulting from low hydraulic conductivity and which can then further develop into RTSs (Jorgenson and Osterkamp, 2005; Lewkowicz, 2007; Lamoureux and Lafreniere, 2009; Lewkowicz and Way, 2019; Jones et al., 2019). RTSs expand upslope due to the continual exposure and melt of ground ice at a headwall, thus mobilizing thawed materials which are transported downslope through the scar zone (Kokelj and Jorgenson, 2013; Zwieback et al., 2020). RTSs can grow where ground ice content and topographic settings allows for a continued instability and removal of thawed soils (Burn and Lewkowicz, 1990; Lacelle et al., 2010; Kokelj and Jorgenson, 2013).*

**RC2:** In the line 39 you state "On 40 a pan-Arctic scale, RTSs have a large variation in size, ranging from small active-layer detachments to large mega slumps with headwalls heights of up to 40m (Kokelj et al., 2015; Murton et al., 2017)." Active layer detachment slides and RTS in some publications are considered to be of different types. Please, check: Active Layer Detachment Slides and Retrogressive Thaw Slumps Susceptibility Mapping for Current and Future Permafrost Distribution, Yukon Alaska Highway Corridor Andre Ìe Blais-Stevens, Marian Kremer, Philip P. Bonnaventure, Sharon L. Smith, Panya Lipovsky and Antoni G. Lewkowicz. However, if you merge all genetically different types of cryogenic landslides in one term "RTS" in this study, then just, please, state it clearly in terminology definition. Please,

check: Leibman, M., Khomutov, A., & Kizyakov, A. (2014). Cryogenic landslides in the Arctic plains of Russia: Classification, mechanisms, and landforms. In *Landslide Science for a Safer Geoenvironment* (pp. 493-497). Springer, Cham.

**AC:** We characterized RTS earlier by the presence of a headwall and refrain from using the term active-layer slides here. We added a reference to Swanson and Nolan, 2018 for small RTSs and changed the sentence to:

*On a pan-Arctic scale, RTSs have a large variation in size, ranging from small slumps with headwall heights of less than a meter up to large mega slumps with headwalls heights of up to 40m (Swanson and Nolan, 2018; Kokelj et al., 2015; Murton et al., 2017).*

**RC2:** *Line 61:* "In this work our goal is to map and investigate RTSs on the northern Taymyr Peninsula, a region that is known to be susceptible to thaw slumping." – please, add the reference.  As you have stated in line 36 the RTS occurrence is linked to the ground ice thawing. In my opinion it is important to mention what kind of ground ice was found in Taymyr that can be "responsible" for RTS occurrence (this will perfectly explain thaw slumping susceptibility).

Massive ground ice thawing was reported to lead to RTS occurrence in the Kara sea region (Belova, 2020; https://www.poac.com/Proceedings/2021/POAC21-005.pdf). The presence of massive ground ice in the Northern Taymyr was mentioned in Massive ground ice database.

Please, have a look at: Streletskaya, I.D., Ukraintseva, N.G. & Drozdov, I.D., 2001. Massive ground ice database. [Online] Available at: http://www.geogr.msu.ru/cafedra/crio/Tabular/

Locations of massive ground ice in the Northern Taymyr: http://www.geogr.msu.ru/cafedra/crio/Tabular/Data/Regions/TAYMIR%20PENISULA%20AND%20LAPTEV%20SEA%20COAST.htm

Moreover, field observations stated in the Geocryology of the USSR, 1989* declare the presence of massive ground ice in Taymyr reaching up to 3 m of thickness as well as the occurrence of ice wedges up to 6 m of depth.

*- Yershov, E.D. (ed.). 1989. Geocryology of the USSR. Central Siberia. Moscow: Nedra, 414 pp. (in Russian). Open access, in Russian: https://www.studmed.ru/ershov-e-d-red-geokriologiya-sssr-srednyaya-sibir_2217c861271.html

**AC:** We added the reference for RTS development and also elaborate on massive ice content adding the refences you suggested:

*In this work our goal is to map and investigate RTSs on the northern Taymyr Peninsula, a region containing massive ground 70 ice, remnant from the Kara Ice-Sheet and which is known to be susceptible to thaw slumping (Grosval'd et al., 1986; Yershov, E.D., 1989; Alexanderson et al., 2002).*

**Study area**

**RC2:** *Line 86:* "During winter the region experiences monthly mean temperatures below -30° C (Matveyeva, 1994)." – this is a bit outdated due to climate warming (also: "air" is missed in the sentence).

For example, mean air temperature from the meteorological station at the Cape Chelyuskin for the coldest winter months of December, January and February was -24° for 2010/2011 and -22° for 2020/2021.

Thus, I'd suggest the following:

Please, define the exact months you were working with since "the winter" in the Siberian Artic starts in ~October and lasts till ~May. It would be great if months will be added at Fig.2 as well.

Create a graph with monthly mean air temperature for the months that were considered in the research.

I am pretty sure that recent in-situ air temperature data will update and enhance current study area description a lot.

You can look at the archived meteorological data freely available for your area at the meteorological station at the Cape Chelyuskin (77° 43' N., 104° 18' E) https://rp5.ru/%D0%90%D1%80%D1%85%D0%B8%D0%B2_%D0%BF%D0%BE%D0%B3%D0%BE%D0%B4%D1%8B_%D0%BD%D0%B0_%D0%BC%D1%8B%D1%81%D0%B5_%D0%A7%D0%B5%D0%BB%D1%8E%D1%81%D0%BA%D0%B8%D0%BD

It is in Russian, but built-in browser translator should help, otherwise I can help.

Meteorological station at the Sterlegov Cape (75° 24' N, 88° 47' E) https://rp5.ru/%D0%90%D1%80%D1%85%D0%B8%D0%B2_%D0%BF%D0%BE%D0%B3%D0%BE%D0%B4%D1%8B_%D0%BD%D0%B0_%D0%BC%D1%8B%D1%81%D0%B5_%D0%A1%D1%82%D0%B5%D1%80%D0%BB%D0%B5%D0%B3%D0%BE%D0%B2%D0%B0

And

*Line 93:* "During the Middle and Late Weichselian (MIS 4 to 2) the ice retreated step-wise and also temporary re-advanced leading to several ice-marginal zones (NTZ) north of the Byrranga Mountains with associated features of glacial, glaciofluvial and glaciolacustrine deposits including buried glacial ice."

This is one of the geological concept stating the presence of the deposits only of the glacial genesis. Meanwhile in the Geocryology of the USSR (page 142, in Russian) it is stated that there are not only glacial but also marine deposits on the Taymyr peninsula.

The following are the sheets of the state geological (quaternary) map for the Northern Taymyr peninsula, where marine deposits also take place. Blue «m» index stands for marine deposits:

State geological map. Scale 1: 1,000,000 (third generation). Taimyr-Severnaya Zemlya series. Sheet S-47 (Taymyr Peninsula), 2015. Map of Quaternary formations. VSEGEI: St. Petersburg.

https://webftp.vsegei.ru/GGK1000/S-47/S-47_KQO_1.pdf

State geological map. Scale 1: 1,000,000 (third generation). Taimyr-Severnaya Zemlya series. Sheet T-45-48 (Taymyr Peninsula), 2013. Map of Quaternary formations. VSEGEI: St. Petersburg.

https://webftp.vsegei.ru/GGK1000/T-45-48/T-45-48_KQO_1.pdf

State geological map. Scale 1: 1,000,000 (third generation). Taimyr-Severnaya Zemlya series. Sheet S-46 (Taymyr Peninsula), 2016. Map of Quaternary formations. VSEGEI: St. Petersburg.

https://webftp.vsegei.ru/GGK1000/S-46/S-46_KQO.pdf

So, please mention this point of view as well.

**AC:** We made the suggested changes to the study region description and added temperature plots to Figure 1 with air temperature data from Cape Chelyuskin and Cape Sterlegov as well as the average air temperature data over the whole study region from ERA5. We added the updated temperature values to the Study region description. We added the information about which month where consider as "winter" (November to April) to the caption of Figure 1 and 2 and to the method section and also note the presence of marine deposits based on the Geocryology of the USSR :

*The climate is characterized as polar Arctic with a mean annual air temperature of about -10 ◦ C. The mean July air temperatures are 2 to 5 ◦ C in the far North and 7 to 10 ◦ C in the South of our study region. During winter the region experiences monthly mean air temperatures below -20 ◦ C*

*(Figure 1b). The area is underlain by continuous cold permafrost, with estimated permafrost temperatures of -11 ◦ C to -8 ◦ C (Obu et al., 2018).*

*Quaternary glaciations and marine transgressions have made a mark on the topography and stratigraphy of the Taymyr Peninsula. The entire peninsula was covered by the Kara Sea Ice Sheet during the Saalian glacial period (140 - 130 ka; MIS 6; Hjort et al. (2002)). Deglaciation during the Eemian interglacial (MIS 5e) was followed by renewed glaciation during the Early Weichselian in MIS 5d to 5a (Hjort et al., 2004; Batchelor et al., 2019). During the Middle and Late Weichselian (MIS 4 to 2) the ice retreated step-wise and also temporary re-advanced leading to several ice-marginal zones (NTZ) north of the Byrranga Mountains with associated features of glacial, glaciofluvial and glaciolacustrine deposits including buried glacial ice. The Geocryology of the USSR also indicate some marine deposits in the region (Proskurnin et al., 2016). Two ice-marginal zones (NTZ 1 and 2) have been associated with the Early and Middle Weichselian and are located close to each other. The most recent of these ice-marginal zones stemmed from the Last Glacial Maximum (NTZ 3) (Möller et al., 2011). The location of the ice-marginal zones is shown in Figure 1.*

**Methods**

**RC2:** *Line* 121: "In the manual inspection of the detection location, as well as in the delineation of the affected area, we additionally used time-series of Sentinel-2 and Planet Rapid-Eye images (Drusch et al., 2012; Planet-Team, 2018). For each RTS we computed the volumetric and area change based on the drawn polygons."

*Regarding the manual identification:* How is the manual identification verified? Unfortunately, in the error calculations no human error of manual identification was implied. For example, in the study of Lewkowicz and Way, 2019, where manual identification also took place, the authors applied 5 iterations to examine the data. If any kind of such verification is not possible to perform now, then, please, at least mention the possible human error in the Discussion.

*Regarding the manual delineation of the affected area:* When looking at your data it is noticeable that some of the polygons are located very close to each other, thus can represent not two separate RTSs (as stated in the manuscript) but two active parts of one RTS. I have compared the polygons 1075 and 1074 that are identified for TP2, in particular between 01.2018 and 01.2021. According to the satellite image available at the Planet for the September 2020 these two polygons lie within one RTS. Thus, it is incorrect to calculate them as two different. I can assume there could be more of such errors in the data because I've randomly checked only few polygons. Please,

check the data for such cases and elaborate this point in your discussion as another human error option.

**AC:**

*Regarding the manual classification*: Unfortunately, we do not have a large number of students available to conduct the RTS mapping multiple times. Generating the polygons once already took several weeks. We tried to estimate the mapping error by increase and decreasing the polygons, but we agree that an approach with multiple mapping persons would be advantages. We highlighted this additional error source in the discussion.

*Regarding the manual delineation of the affected area:* We agree that it can be difficult so separate RTS that are close to each other. Furthermore, RTS can merge over time. This can introduce an additional error source which we highlighted again in the discussion. In our case we used the DEM difference data to separate RTSs, not the optical/infrared observations that you showed. In optical/infrared data only the induced vegetation changes are visible and it is not clear if the headwall is connected or if they are just connected by the debris outflow. In the two RTSs mentioned, RTS 1074 is at an about 10m lower elevation than 1075, indicated by the generated DEMs (Supplement Figure 1). An about 30-50m gab between significant elevation losses is visible in the DEM difference images (Supplement Figure 2). We think that these RTS can indeed be seen as two separate ones and only the thawed soil is transported along the same path (Supplement Figure 3). We agree that for these RTSs the separation is not obvious and we highlighted these difference in the discussion e.g. they could also be connected but with small headwall heights between them, such that it is not visible in DEM difference images. This should especially be considered when this dataset is compared to optical/infrared generated RTS inventories.

*Regarding mapping errors, we tried to estimate the error in the area change by increasing and decreasing the size of the drawn polygons. Here additional human errors in drawing the polygons are not considered. An approach taken by previous studies* (e.g. Lewkowicz andWay (2019)) *were polygons were drawn multiple times by different trained persons was not feasible. Furthermore, relating the changes to individual RTSs becomes difficult for RTSs in close proximities and due to RTS coalescence. In this study we separated RTSs based on the induced elevation changes. RTSs that seem connected based on the induced vegetation changes obtained from optical and infrared observations could thus be related to multiple RTSs in our RTS inventory if they are only connected by small RTS headwalls or by the flow of the thawed soil downwards.*

**RC2:** *Line* 178: "Additionally to the SOC, the massive ice content needs to be known. Here the data availability is even more scars and uncertain than for the SOC content."

Ice content for Taymyr is more or less covered in the Geocryology of USSR by Yershov, 1989 (pages 145, 146, in Russian): it ranges from 30% to 70%, which is nicely in line with your initial assumptions, that is worthy to mention.

**AC:** We added the reference and change this part to:

*Here the data availability is more scars and uncertain* than it is for the SOC content*. Yershov E.D. (1989) reported massive ice-contents on the Taymyr* Peninsula *in the range from 30% to 70%. Past studies that estimated massive ice content at RTS locations have found values in a wide range.*

**Results**

**RC2:** *Line* 298: "during 2020 related to a Siberian heatwave (Overland and Wang, 2021)"

Please, move this reference to the first mention of "Siberian heatwave" - somewhere to section 3.5

**AC:** We did not mention specifically the "Siberian heatwave" in section 3.5, but first at the end of the Introduction, where name the reference already. We removed the reference here.

**Discussion**

**RC2:** *Line* 314: "This study is the first that identified such an initiation event also for RTSs in the north-Siberian Arctic."

This is not correct. Field observations by Khomutov et al., 2017 demonstrated the activation of different cryogenic landslides and thermocirques (RTS) initiation in Yamal linked to the extremely warm summer in 2012. Please, correct accordingly: Khomutov, A., Leibman, M., Dvornikov, Y., Gubarkov, A., Mullanurov, D., & Khairullin, R. (2017, May). Activation of cryogenic earth flows and formation of thermocirques on central Yamal as a result of climate fluctuations. In Workshop on World Landslide Forum (pp. 209-216). Springer, Cham.

**AC:** Thank you for pointing out this reference. We changed this part accordingly:

*Mass initiations of RTSs following extreme summer temperatures have been documented on the Canadian Arctic Archipelagos (Jones et al., 2019; Lewkowicz and Way, 2019) and the Yamal Peninsula in Siberia (Khomutov et*

*al., 2017). This study is the first that identified such an initiation event for RTSs in the Northern Taymyr Peninsula.*

**Technical corrections**

**RC2:**

Abstract Line 3: "carbon mobilization on the Taymyr Peninsula" -> "carbon mobilization on the Northern Taymyr Peninsula"

Comment: to be consistent it is better to use the capital letter "N" in "the Northern Taymyr" everywhere. Since the study was for the northern part of the peninsula, thus it should be stated as "Northern" everywhere related to the study area.

*Line* 61: northern Taymyr -> Northern Taymyr.

*Line* 77: northern Taymyr -> Northern Taymyr.

Figure 1 caption: northern Taymyr Peninsula -> Northern Taymyr Peninsula.

Figure 1 caption: northern Taymyr Ice-Marginal Zones (NTZ 1-3) -> Northern Taymyr Ice-Marginal Zones (NTZ 1-3).

*Line* 226: northern Taymyr Peninsula -> Northern Taymyr Peninsula.

*Line* 235: Taymyr Ice-Marginal Zone -> Northern Taymyr Ice-Marginal Zone

Heading of the section 5.1 "Acceleration of RTS activity on the Taymyr Peninsula" -> "Acceleration of RTS activity on the Northern Taymyr Peninsula"

Heading of the section 5.3 "Acceleration of RTS activity on the Taymyr Peninsula" -> "Acceleration of RTS activity on the northern Taymyr Peninsula"

*Line* 382: "carbon on the Taymyr Peninsula " -> "carbon on the northern Taymyr Peninsula"

**AC:** We added "northern" to the abstract line 3. To be consistent, we checked each occurance and changed it to "Northern Taymyr Peninsula".

**RC2:** *Line* 93: Please, add the definition of "NTZ" acronym in the text as well. Now it is described only in the figure caption.

**AC:** We added the definition

**RC2:** *Line* 132: Please, define elevation error sign.

**AC:** The elevation error described here indicates the standard deviation from the measured height thus + and – the measured elevation..

**RC2:** *Line* 193: Please, add the definition of "resp." abbreviation.

**AC:** We remove the abbreviation and wrote the full word "respectively".

**RC2:** *Line* 247: You probably meant colon ":" not a semicolon ";", right? Otherwise, it seems to be a bit complicated to read.

Fig.7 caption: show -> showS; are -> area.

Fig. 8 caption: the description for "b)" is missing.

The caption of Fig. 11: The marker location correspond to -> corresponds to.

**AC:** Corrected.

[Figure]

*Figure 1: Left: DEM generated in winter 2017/18, Right: DEM generated in Winter 2010/21*

[Figure]

*Figure 2: Left: DEM difference between winter 2017/18 and winter 2020/21, Right: Test (Significant) Score using a 5x5 moving window. This takes into account the the errors in the elevation measurements (see Bernhard et al. 2020 for a detailed explanation of the Test-Score).*

[Figure]

*Figure 3: Left: Sentinel-2 False color image taken on 28.08.2017 (before the first dem), Right: Sentinel-2 False color image taken on 23.09.2020 (end of summer before second DEM)*